

# The oldest record of the Steller sea lion *Eumetopias jubatus* (Schreber, 1776) from the early Pleistocene of the North Pacific

Nahoko Tsuzuku[1] and Naoki Kohno[1,2]

[1] Department of Life and Environmental Sciences, University of Tsukuba, Tsukuba City, Ibaraki Prefecture, Japan
[2] Department of Geology and Paleontology, National Museum of Nature and Science, Tsukuba City, Ibaraki Prefecture, Japan

Corresponding author
Nahoko Tsuzuku,
tsuzuku@geol.tsukuba.ac.jp

## ABSTRACT

The extant genera of fur seals and sea lions of the family Otariidae (Carnivora: Pinnipedia) are thought to have emerged in the Pliocene or the early Pleistocene in the North Pacific. Among them, the Steller sea lion (*Eumetopias jubatus*) is the largest and distributed both in the western and eastern North Pacific. In contrast to the limited distribution of the current population around the Japanese Islands that is now only along the coast of Hokkaido, their fossil records have been known from the middle and late Pleistocene of Honshu Island. One such important fossil specimen has been recorded from the upper lower Pleistocene Omma Formation (ca. 1.36–0.83 Ma) in Kanazawa, Ishikawa Prefecture, Japan, which now bears the institutional number GKZ-N 00001. Because GKZ-N 00001 is the earliest fossil having been identified as a species of the sea lion genus *Eumetopias*, it is of importance to elucidate the evolutionary history of that genus. The morphometric comparisons were made among 51 mandibles of fur seals and sea lions with GKZ-N 00001. As results of bivariate analyses and PCA based on 39 measurements for external morphologies with internal structures by CT scan data, there is almost no difference between GKZ-N 00001 and extant male individuals of *E. jubatus*. In this regard, GKZ-N 00001 is identified specifically as the Steller sea lion *E. jubatus*. Consequently, it is recognized as the oldest Steller sea lion in the North Pacific. About 0.8 Ma, the distribution of the Steller sea lion had been already established at least in the Japan Sea side of the western North Pacific.

## INTRODUCTION

The extant pinnipeds of the order Carnivora are divided into three families: the Otariidae, the Odobenidae, and the Phocidae (*Berta, Churchill & Boessenecker, 2018*). Among them, the otariids are distributed mostly in the Pacific Ocean and composed of fur seals and sea lions. Sea lions are generally large in body size; in particular, the Steller sea lion (*Eumetopias jubatus*) in the North Pacific is the largest among them. In contrast to the relatively limited distribution of the current population of the Steller sea lion in Japan, which is now only around Hokkaido (*Loughlin, Rugh & Fiscus, 1984*;

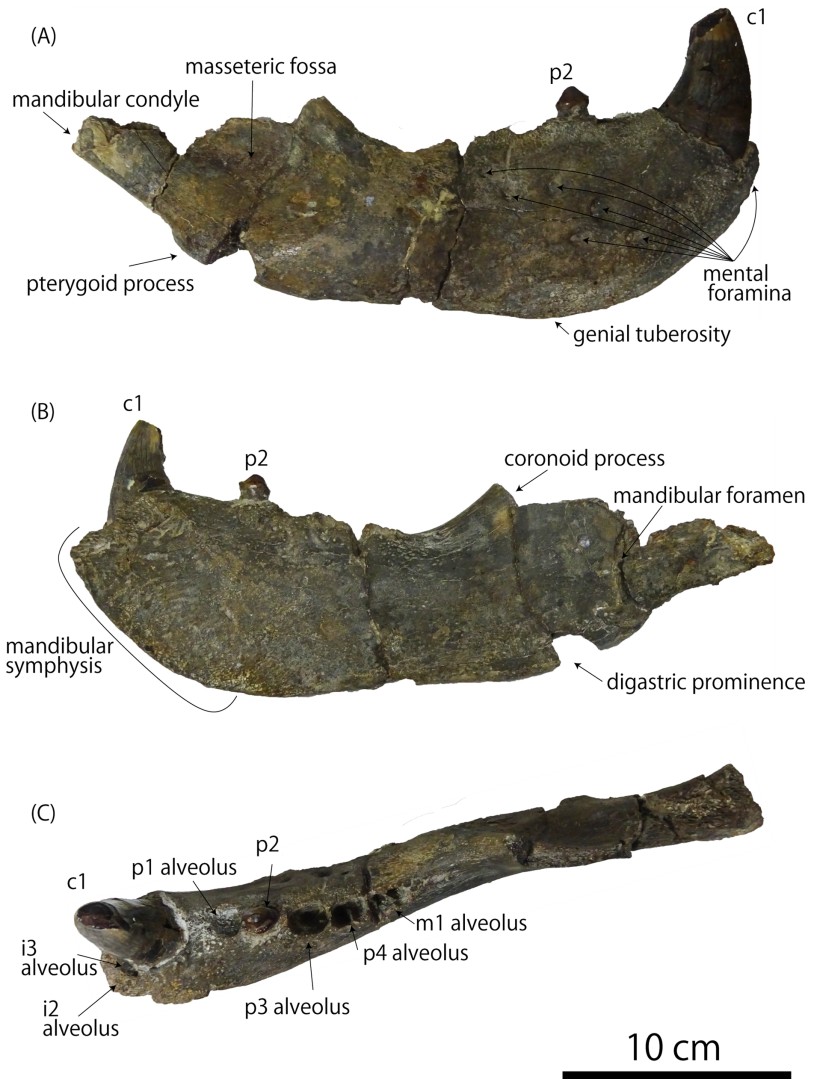

**Figure 1** **Right mandibular fossil of *Eumetopias jubatus* (GKZ-N 00001).** (A) Lateral aspect (B) medial aspect (C) dorsal aspect. Photo credit: Nahoko Tsuzuku.

*Loughlin, Perez & Merrick, 1987*), their distribution during the Pleistocene is thought to be relatively broad, because their mandibular and tooth fossils have been known from the Pleistocene of Honshu Island. One such important fossil specimens has been recorded from the upper lower Pleistocene Omma Formation (ca. 1.36–0.83 Ma) in Kanazawa, Ishikawa Prefecture, Japan, which bears the institutional number GKZ-N 00001 (Figs. 1 and 2; *Kaseno, 1951*; *Shikama, 1953*; *Mitchell, 1968*).

In October 1946, a right mandibular fossil (i.e., GKZ-N 00001) was collected by the late Dr. Yoshio Kaseno (then Kanazawa University) and third year students of the Biological Department, Kanazawa Higher Normal School, Ishikawa Prefecture, Japan. The specimen came from the upper part of the Omma Formation at a road cutting south of Gosyo village (now Gosyo Town), northeastern end of Kanazawa City, Ishikawa Prefecture, central Japan. While *Kaseno (1951)* identified GKZ-N 00001 tentatively as a

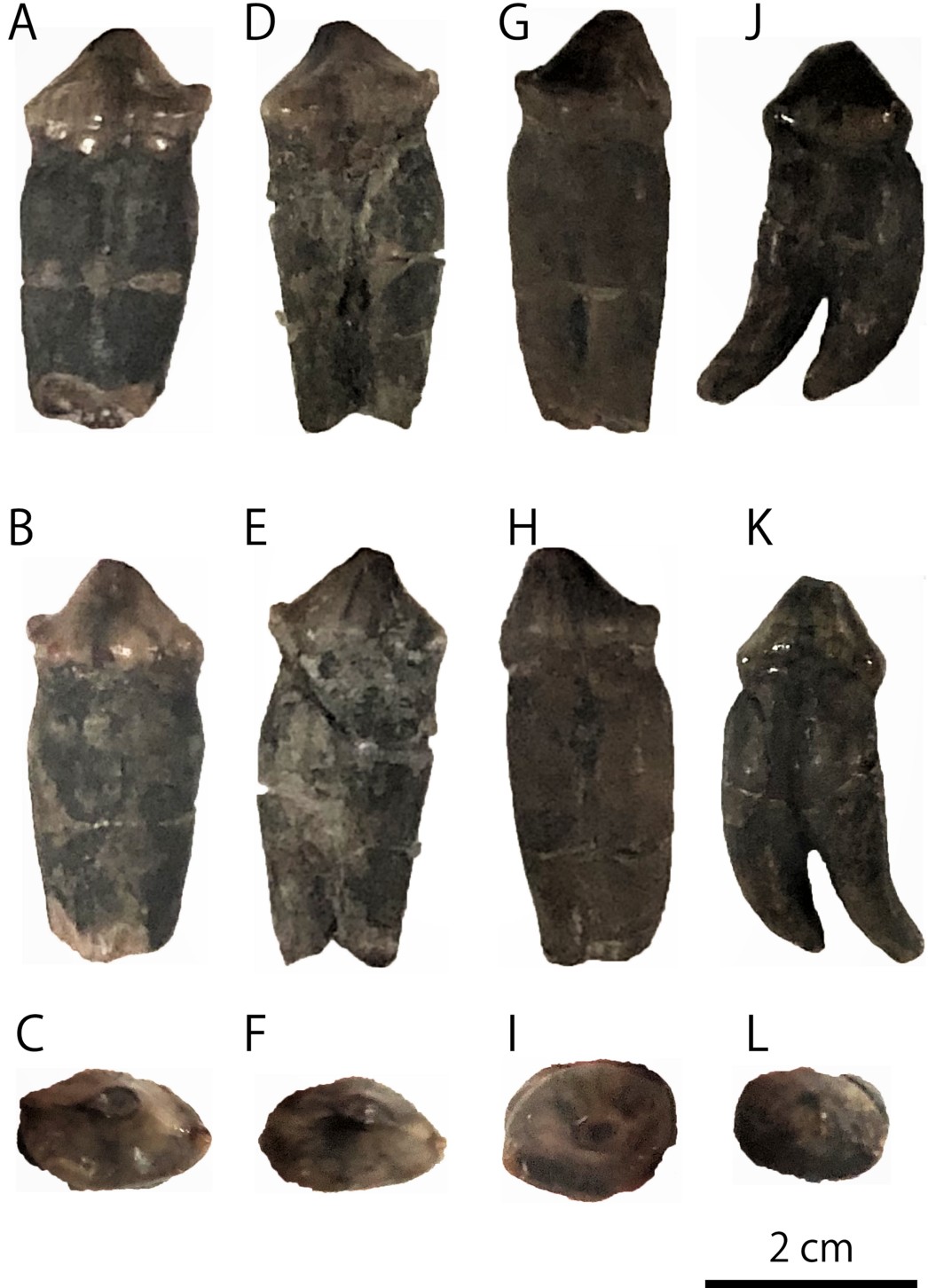

**Figure 2 Detached molariform teeth of GKZ-N 00001.** (A) Lingual view of left p4 (B) buccal view of left p4 (C) occlusal view of left p4 (D) lingual view of right p4 (E) buccal view of right p4 (F) occlusal view of right p4 (G) lingual view of left P4 (H) buccal view of left P4 (I) occlusal view of left P4 (J) lingual view of right M1 (K) buccal view of right M1 (L) occlusal view of right M1. Photo credit: Nahoko Tsuzuku.

**Table 1** Sample list.

| Scientific name | Sex | N | Registration number |
|---|---|---|---|
| *Eumetopias jubatus* | Male | 12 | NMNS-KK14, 15, 23, 51, 63, 69, 73, 167, 169, 192, NSMT-M5627, 28387 |
| | Female | 12 | NMNS-KK11, 53, 54, 55, 56, 139, 146, 154, 158, 165, 166, NSMT-M17123 |
| *Callorhinus ursinus* | Male | 3 | NSMT-M2454, 46874, 17140 |
| | Female | 10 | NMNS-KK05, 08, 10, 22, 24, 141, 151, NSMT-M42128, 35148, 1995 |
| *Zalophus japonicus* | Male | 10 | HM-55953-18-1~8/DCIFC-ER11H,-HM2L |
| | Female | 1 | DCIFC-HM2 · 97R.No.30262 |
| *Proterozetes ulysses* | Male | 2 | USNM 187109 (*Barnes, Ray & Koretsky, 2006*), UCMP 219377 (*Poust & Boessenecker, 2017*) |

species of "*Allodesmus*" which is an extinct pinniped known from the Miocene of the North Pacific, *Shikama (1953)* pointed out that GKZ-N 00001 might belong to *E. jubatus* (*E. jubata* at that time). Later, *Mitchell (1968)* also suggested that GKZ-N 00001 be studied further before being unequivocally identified as conspecific with *E. jubatus*. Currently, the specific identification of GKZ-N 00001 is still ambiguous and in controversy (*Kohno & Tomida, 1993*; *Barnes, Ray & Koretsky, 2006*) and has not been confirmed yet.

At the time when GKZ-N 00001 was found, the Omma Formation had been considered to be Pliocene in age, but it turned out to be early Pleistocene (ca. 1.36–0.83 Ma) based on the calcareous microfossil stratigraphy (*Takayama et al., 1988*). The extant genera in the Otariidae including *Eumetopias* are considered to be branched off at the end of the Pliocene (*Repenning & Tedford, 1977*), and the oldest fossil record of extant otariid genera is known from the late Pliocene to the early Pleistocene (*Berta & Deméré, 1986*; *Kohno & Yanagisawa, 1997*). These studies suggest that the early Pleistocene that may also correspond to the emergence of the genus *Eumetopias* is important time to elucidate the evolutionary history of the otariids. Therefore, highlighting the meaningfulness of proper identification of GKZ-N 00001 is meaningful.

## MATERIALS AND METHODS

In order to compare fossil specimens with modern taxa, the morphometric analyses are performed using 51 mandibles of fur seals and sea lions: 12 male and 12 female individuals of the Steller sea lion *E. jubatus*, three male and 10 female individuals of the Northern fur seal *Callorhinus ursinus*, 10 male and one female individuals of the Japanese sea lion *Zalophus japonicus*, two male individuals of the Odysseus sea lion *Proterozetes ulysses*, and GKZ-N 00001 (Table 1; Data S1). A total of 21 landmarks are defined with reference to the previous morphometric research (*Berta & Deméré, 1986*; *Isono, 1998*; *Adam & Berta, 2002*; *Brunner, 2004*; *Boessenecker, 2011*; *Kienle & Berta, 2015*), and a total of 39 measurements are taken using a digital caliper to the nearest 0.01 mm (Fig. 3; Tables 2 and 3; Table S1).

Based on these measurements, bivariate analyses and principal component analysis (PCA) are performed. PCA is implemented in R 3.5.1 (*R Core Team, 2018*). In addition, the mandibles are observed with the micro-computed tomographic scanner using Microfocus CT, TXS320-ACTIS at the National Museum of Nature and Science, Tokyo, Japan.

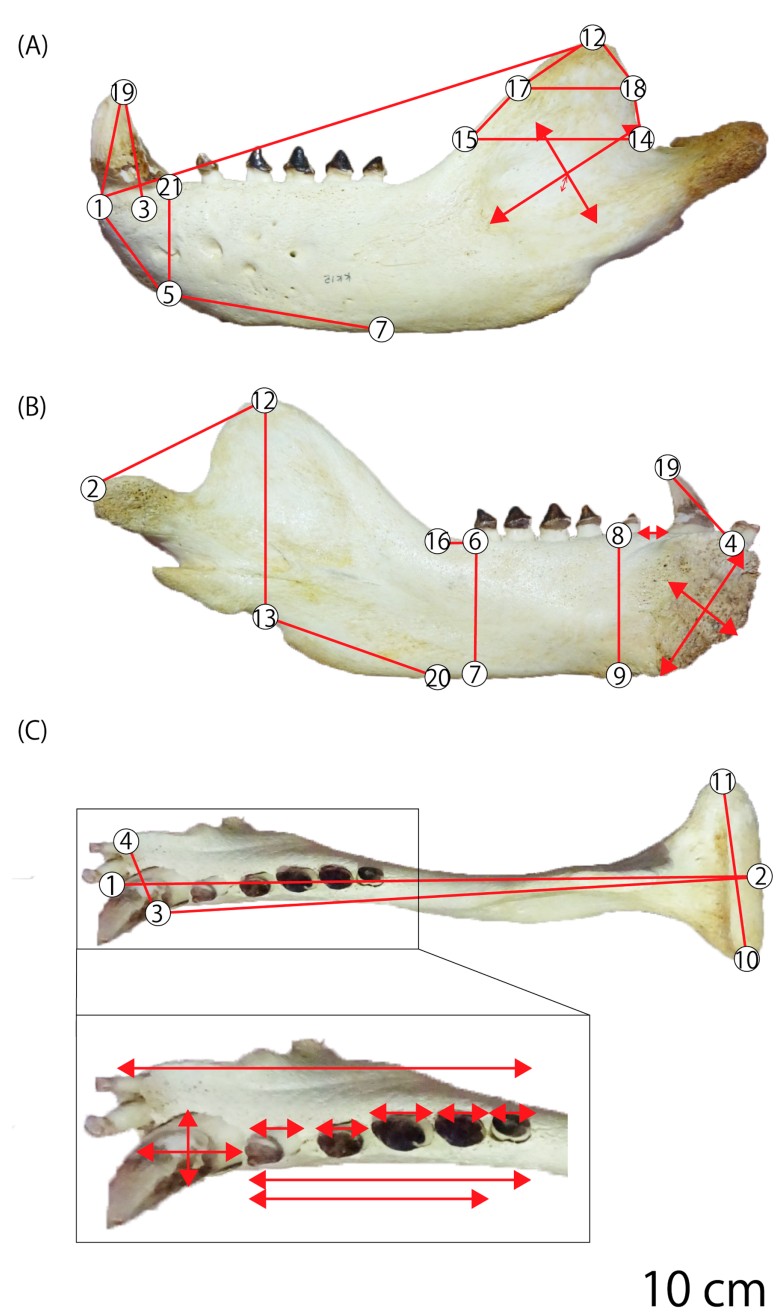

**Figure 3 Mandibular landmarks and measurements used in this study.** (A) Lateral aspect (B) medial aspect (C) dorsal aspect. Photo credit: Nahoko Tsuzuku.

## SYSTEMATIC PALEONTOLOGY

MAMMALIA *Linnaeus, 1758*
CARNIVORA *Bowditch, 1821*
PINNIPEDIA *Illiger, 1811*
OTARIIDAE *Gill, 1866*

**Table 2 Mandibular landmarks used in this study.**

| | |
|---|---|
| 1 | Rostral tip of mandible (gnathion) |
| 2 | Caudal-most point of mandibular condyle |
| 3 | Lateral edge of canine midpoint |
| 4 | Medial edge of mandibular symphysis caudal to the first incisor |
| 5 | Ventral edge of mandible underneath the canine |
| 6 | Lateral and caudal edge of last postcanine |
| 7 | Ventral edge of mandible underneath the last postcanine |
| 8 | Lateral and caudal edge of p1 |
| 9 | Ventral edge of mandible underneath the p1 |
| 10 | Lateral-most point of mandibular condyle |
| 11 | Medial-most point of mandibular condyle |
| 12 | Dorsal-most point of coronoid process (koronion) |
| 13 | Ventral edge of mandible underneath the tip of the coronoid process |
| 14 | Caudal-most point of coronoid process |
| 15 | Rostral point horizontal to caudal-most point of coronoid process |
| 16 | Rostral start point of the coronoid process |
| 17 | Rostral-most point on curving edge of coronoid process |
| 18 | Caudal-most point on curving edge of coronoid process |
| 19 | Dorsal tip of canine |
| 20 | Ventral edge of mandible underneath the start of the coronoid process |
| 21 | Lateral and caudal edge of canine |

**Included Genera:** [extant genera] *Arctocephalus Cuvier, 1826*; *Arctophoca Peters, 1866*; *Callorhinus Gray, 1859*; *Eumetopias Gill, 1866*; *Neophoca Gray, 1866*; *Otaria Peron, 1816*; *Phocarctos Peters, 1866*; *Zalophus Gill, 1866*. [extinct genera] *Eotaria Boessenecker & Churchill, 2015*; *Pithanotaria Kellogg, 1925*; *Thalassoleon Repenning & Tedford, 1977*; *Hydrarctos* (*De Muizon, 1978*) *Berta & Deméré, 1986*; *Proterozetes Barnes, Ray & Koretsky, 2006* (after *Berta & Churchill, 2011*; *Velez-Juarbe, 2017*).

**Emended Diagnosis of Family:** [Crown Otariidae (*C. ursinus*, northern sea lion clade, and southern otariid clade)] single rooted P3, P4, p2, and p4. Pronator teres insertion positioned on the proximal 40% of the radius. Secondary shelf of the sustentaculum of the calcaneum developed as a wider shelf (modified from *Churchill, Boessenecker & Clementz, 2014*).

*EUMETOPIAS Gill, 1866*

**Included Species:** Only the extant species *Eumetopias jubatus* (*Schreber, 1776*).

**Remarks:** *Horikawa (1981)* described a partial skeleton including some isolated cheek teeth as the holotype of a new species in the genus, i.e., *Eumetopias ojiyaensis*, based on comparisons mainly of the upper incisor and canine with a female of *E. jubatus*. However, the characters he mentioned (e.g., overall similarity of teeth, condition of accessory

**Table 3 Measurements (in mm) of GKZ-N 00001 for morphometric analyses.** The numbers in parentheses correspond to each landmark.

| | |
|---|---|
| Rostral tip of mandible–ventral edge of mandible underneath the canine (1–5) | 50.35 |
| Width of mandible (3–4) | 42.76 |
| Depth of ramus at c1 (5–21) | 75.15 |
| Depth of ramus at p1 (8–9) | 76.46 |
| Depth of ramus at m1 (6–7) | 66.34 |
| Ventral edge of mandible underneath the canine–ventral edge of mandible underneath the last postcanine (5–7) | 100.13 |
| Lateral and caudal edge of last postcanine–rostral start point of the coronoid process (6–16) | 27.60 |
| Major axis of mandibular symphysis | 81.31 |
| Minor axis of mandibular symphysis | 50.29 |
| Major axis of masseteric fossa | 68.43 |
| Minor axis of masseteric fossa | 45.73 |
| Depth of masseteric fossa | 13.18 |
| Tooth length | 123.89 |
| Cheek tooth length | 84.03 |
| Transverse width of c1 | 27.32 |
| Mesiodistal diameter of c1 | 35.00 |
| Anteroposterior length of p1 | 13.33 |
| Anteroposterior length of p2 | 12.79 |
| Anteroposterior length of p3 | 16.07 |
| Anteroposterior length of p4 | 14.51 |
| Anteroposterior length of m1 | 18.42 |
| Diastema length between c1 and p1 | 10.04 |

cusps on the molar etc.) are plesiomorphic or nondiagnostic for the sea lions and unavoidable the potential identification of the holotype to a species of other sea lion genera within the Otariidae because of its incompleteness. We consider *E. ojiyaensis* a *nomen dubium* and suggest a temporary pending on both generic and specific identification for its holotype until much better specimens at least including the mandible will be obtained from the type locality or the same formation.

**Diagnosis of Genus:** As for the species.
*Eumetopias jubatus* (*Schreber, 1776*)

**Emended Diagnosis:** *Eumetopias jubatus* is the largest species in body size (about 3.3 m in males) among the family Otariidae. The upper and lower canines are large in males, particularly at the apical ends of roots that are almost evergrowing (less so in females). The mandible is long and the angle between the horizontal and ascending rami is large (about 130–140 degree) in contrast to the smaller angle of less than 130 degree in other genera and species. The masseteric fossa is deep and long, especially in older individuals. Postcanines are unicuspid with well-developed labial and lingual cusples at the base (modified from *Brunner, 2004*).

**Dental formula:** 3/2, 1/1, 4/4, 1/1.

**Referred specimen:** GKZ-N 00001, incomplete right dentary with right lower canine, left p2, left and right p4, left P4 and right M1; collected by the late Yoshio Kaseno (then Kanazawa University) and third year students of the Biological Department, Kanazawa Higher Normal School, in October 1946. Now this specimen is stored at the Graduate School of Natural Science and Technology, Kanazawa University (GKZ).

**Locality of referred specimen:** GKZ-N 00001 was found at a road cutting south of Gosyo Town, north-eastern end of Kanazawa City, Ishikawa Prefecture, Japan. The geographical coordinate is 36°34′51″ North Latitude, 136°40′59″ East Longitude. Currently, no outcrop remains there.

**Formation and Age:** GKZ-N 00001 was yielded from the upper part of the Omma Formation. It is consisted of homogeneous silty fine-grained sandstone in bluish color, from which many fragmentary remains of mollusks such as *Acila*, *Pecten*, *Venericardia*, *Myodora*, *Cardium*, and *Diplodonta* have been collected (*Kaseno, 1951*). The geologic age of the Omma Formation corresponds to the late early Pleistocene (ca. 1.36–0.83 Ma) based on the calcareous microfossil stratigraphy (*Takayama et al., 1988*). The depositional environment of the Omma Formation is thought to be cold temperate water within 0–30 m in depth, on a shoreface or inner shelf, the upper part of the upper shallow-sea zone, on the basis of the habitat preferences of molluskan fossils collected from a bed slightly higher than the horizon of GKZ-N 00001 (*Tsuzuku, 2018*).

**Associated Mammalian Fossils:** *Matsuura (1996)* reported some metacarpals and phalanges of possibly otariid pinnipeds. In addition, many cetacean and some sirenian fossils have also been found from the same formation (*Matsuura & Nagasawa, 2000*), although these have not yet been described in detail.

### Description

**Mandible** (Fig. 1): GKZ-N 00001 is a right mandible consisting of almost complete horizontal ramus and broken coronoid process. The c1 and p2 are in place on the horizontal ramus. Both left and right p4 are also preserved as isolated teeth. In addition, isolated left P4 and right M1 are also preserved, suggesting that the skull as well as the left mandible might also be preserved at the time of its discovery. Based on the robustness of the mandible and the size of the canine relative to the length of the cheek toothrow that reaches 42%, GKZ-N 00001 is definitely a male. The horizontal ramus is long and thick, and has unparallel dorsal and ventral margins. The anterior border of the horizontal ramus (the portion anterior to the incisors and the canine) is expanded anterolaterally. The bone surface of the ramus is rough, but not vascularized. The mandibular symphysis is unfused, elliptical in shape, and upturned anteriorly. The posterior border of the symphysis extends below the anterior margin of p2. The genial tuberosity is very small and located below p3. There are eight distinct mental foramina on the lateral surface of the horizontal ramus. They are nearly rounded or ellipsoidal in form, with the diameter varying from about 3 to 15 mm. The anterior mental foramen is located beneath i3 on the

anterior margin of the horizontal ramus and is slit-like. Its length is about 15 mm and width is 4 mm. The middle mental foramina are located between p1 and p3 on the mid portion of the horizontal ramus, and the most anterior elliptical and large hole is located beneath p1 with the diameter of 11 mm in major axis and 6 mm in minor axis. The second middle mental foramen is located beneath p1 and p2 and rounded in outline, with a diameter of 11 mm. The third middle mental foramen is located beneath p2 on the mid portion of the horizontal ramus and is very large ellipsoidal in form. Its major axis is about 11 mm and minor is 6 mm. The fourth middle mental foramen is located between p2 and p3, and is rounded in outline and 6 mm in diameter. The fifth middle mental foramen is located between p3 and p4 and is rounded in outline and 7 mm in diameter. The sixth middle mental foramen is located beneath p4 and is the smallest rounded hole and 6 mm in diameter. The posterior mental foramen is located beneath the posterior margin of m1 and is the rounded in outline and 9 mm in diameter. Almost all of them are directed anterodorsally, but the fifth and sixth forward mental foramina are directed posterodorsally. Most of the coronoid process is broken away. The ventral margin of the masseteric fossa is preserved on the lateral surface and is relatively deep and large in area. It is anteroposteriorly broad at the base of the coronoid process. The pterygoid process is broken at the ventral margin on the medial surface of the ascending ramus. The mandibular foramen is large, with a diameter of 7 mm. The digastric prominence is very weak. The mandibular condyle is broken away, but the breakage suggests that it was elevated slightly high above the level of the cheek toothrow. The dentary has two incisors, one canine, four premolars and one molar. The dentition converges medially from p1 until p3 and diverges laterally from p4 until m1. Each tooth is moderately spaced, and the diastema between the c1 and p1 is slightly wider than others.

**Teeth** (Figs. 1 and 2): The i2 and i3 are missing, but their alveoli are preserved anterior to the c1. Both are single rooted. The c1 is robust and conical. Its apex is abraded. The pulp cavity of c1 is widely opened. The p1 is missing, but its alveolus is preserved on the dentary. It is single rooted. The p2 is preserved in place and lanceolate in form with sharp cutting edge. It has a highly developed lingual cingulum, and a well developed accessory cusp is located mesially at the base of the crown. The p3 is fallen away, but its alveolus is preserved, which is single rooted and bilobed in outline. The left and right p4 are preserved as detached isolated teeth. The crown is lanceolate with blunt cutting edge. It has poorly developed lingual cingulum with well developed accessory cusp. Their roots are bilobate and vertical to the long axis of the crown. The m1 is missing, but its alveolus indicates that the m1 is apically double rooted.

The detached isolated P4 is preserved. The crown is tall and conical, and bluntly pointed at the tip with single cusp. There is no sharp cutting edge with a cingulum at the base of the crown. There is no well developed accessory cusp. It is bilobate single rooted, and its root is oblique to the long axis of the crown. The M1 is also preserved as a detached tooth. It has conical crown, blunt cutting edge, undeveloped cingulum, and slightly developed accessory cusp at the base of crown. It is apically double rooted and oblique to the long axis of the crown. These roots are curved inward and forward strongly.

## RESULTS OF MORPHOMETRIC ANALYSES

### Bivariate analyses

GKZ-N 00001 is almost as large as the mandible of male Steller sea lion and larger than that of other known sea lions including the recently extinct Japanese sea lion and the middle Pleistocene Odysseus sea lion (Tables 2 and 3; Table S1). In addition to the differences of their absolute sizes, the p1 of the Odysseus sea lion is extremely smaller than other cheek teeth on the mandible (*Poust & Boessenecker, 2017*), and the consequent gradient of cheek tooth sizes against the cheek toothrow length are quite different to that of GKZ-N 00001 and of the Steller sea lion. All these results rule out that GKZ-N 00001 belongs to the Odysseus and/or Japanese sea lions in size of the mandible and to the Odysseus sea lion in proportion of the cheek teeth. Given these differences, a taxonomic definition for GKZ-N 00001 as a species of *Eumetopias* is at least warranted. However, given the high amount of morphological variation within species of sea lions necessitates additional morphological approach to classify GKZ-N 00001 more clearly. For these reasons, we made morphometric analyses for GKZ-N 00001 with *E. jubatus* and at least all the North Pacific sea lions including extinct taxa mentioned above.

We performed 39 measurements among 21 landmarks on each mandible of 51 individuals from GKZ-N 00001 and extant taxa including 13 Northern fur seals and 24 Steller sea lions and extinct taxa including 11 Japanese sea lions and two Odysseus sea lions in total (Fig. 3; Tables 2 and 3; Table S1). These variables were correlated heuristically to each other. Then, highly correlated bivariates were considered to evaluate taxonomic significance for the mandibular fossil. As the results of the analyses, three bivariates by total of six parameters were distinctive among species; the depth of the horizontal ramus at c1 vs the mesiodistal diameter of c1, the depth of the horizontal ramus at p1 vs that of the same portion at m1, and the major axis of the mandibular symphysis vs the minor axis of the same portion (Fig. 4). They were also distinctively differentiated between males and females. In all bivariates, GKZ-N 00001 was plotted slightly larger than the 95% confidence interval of the Ordinary Least Square regression lines of the sampled male individuals of *E. jubatus*.

These results suggest that the difference between GKZ-N 00001 and the male *E. jubatus* is only a little and that the former could be included in the variation of the latter. In fact, the *E. jubatus* samples used in this study are individuals collected from around Hokkaido for a purpose to prevent damage to the fisheries, so the individuals tend to be slightly smaller in the average of their body size than that of the original population. For this reason, GKZ-N 00001 that is plotted slightly larger than the 95% confidence interval as *E. jubatus* could be interpreted as a large male individual of *E. jubatus*.

### Proportion of the canine root

Regarding GKZ-N 00001, the greatest mesiodistal and buccolingual diameters of the lower canine root measured from CT image is 50.3 mm and 27.6 mm respectively (Fig. 5; Table 4). Accordingly, the ratio of the buccolingual to mesiodistal diameter expressed as a percentage is 54.9. *Kohno & Tomida (1993)* suggested that the same ratio of the lower

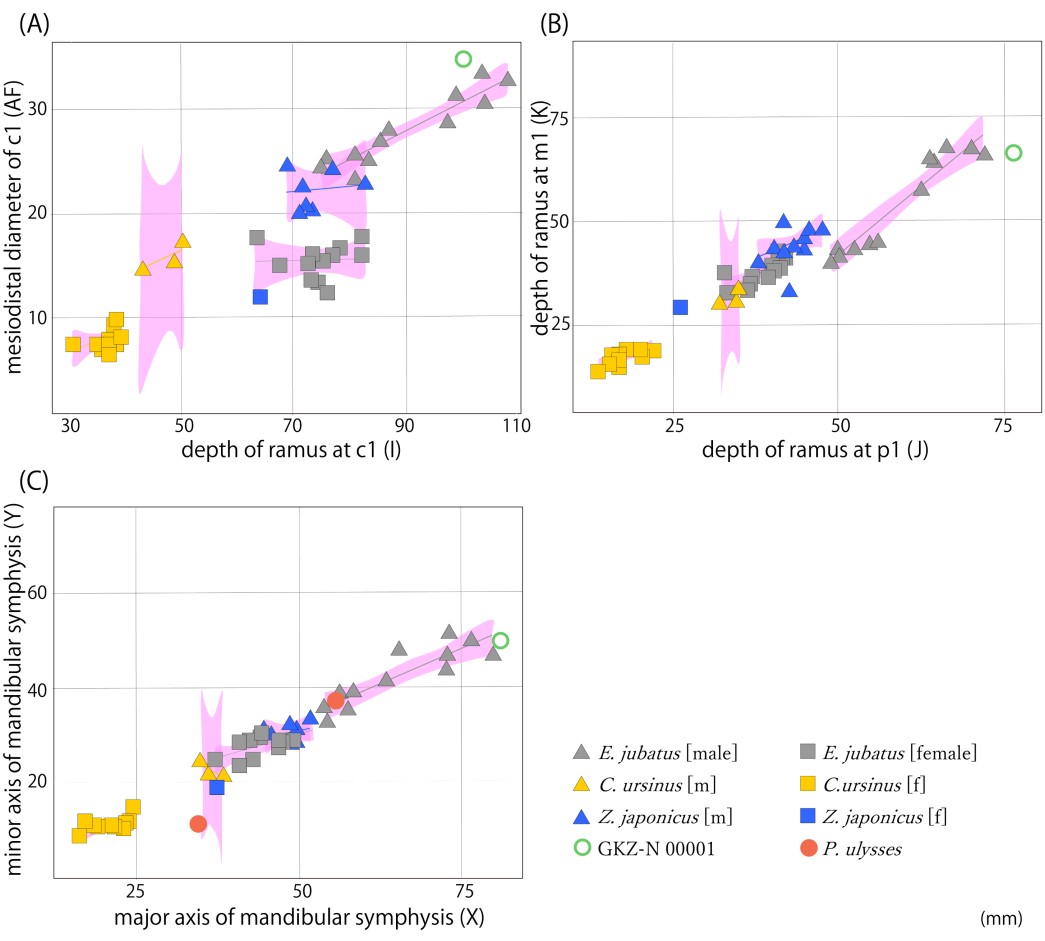

**Figure 4 Result of bivariate analyses.** (A) Depth of horizontal ramus at c1 (I) vs the mesiodistal diameter of c1 (AF); (B) depth of horizontal ramus at p1 (J) vs that of same portion at m1 (K); (C) major axis of mandibular symphysis (X) vs minor axis of same portion (Y). Pink polygons show 95% confidence intervals of the regression lines. Characters in parentheses correspond to measurement points (Table S1).               

canines expressed as a percentage for the males of *E. jubatus* ranged from 51.6 to 61.8 (the mean is 56.6; Table 4). Therefore, GKZ-N 00001 is included within the range of male *E. jubatus* in regard to the canine proportion.

## Principal component analysis

The measurement data from mandibles of GKZ-N 00001 with male and female individuals of *C. ursinus*, *Z. japonicus*, and *E. jubatus* were analyzed using PCA (Fig. 6). As a result of the analysis, the first principal component (PC1) explains the majority of the variation (~88.8%), which represents variation of overall size. The larger individuals such as GKZ-N 00001 and male *E. jubatus* have negative scores, while smaller individuals such as female *C. ursinus* have positive scores. Sexual dimorphism can be observed in each taxon. There is only a little difference between GKZ-N 00001 and the sampled male individuals of *E. jubatus*. The second principal component (PC2) represents the proportion of the dorsoventral height and anteroposterior length of the mandible, meaning that the lower
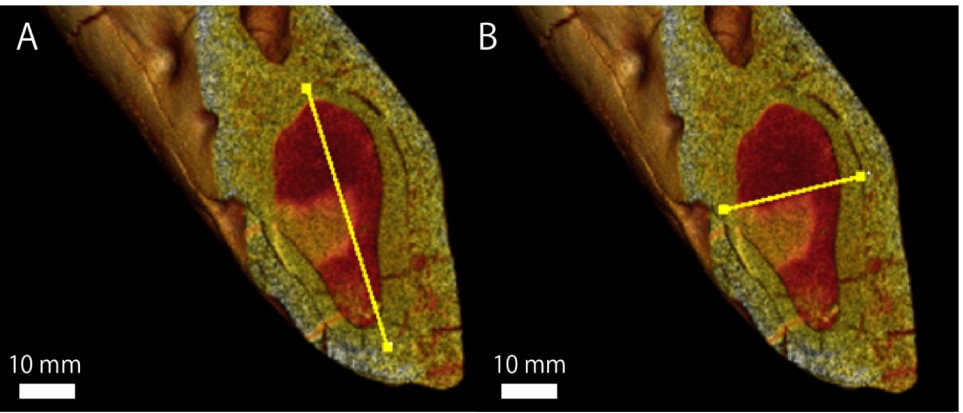

**Figure 5 A micro CT image, showing the maximum length and width of lower canine root of GKZN 00001.** Yellow lines indicate: (A) mesiodistal diameter (B) buccolingual diameter. Photo credit: Naoki Kohno.

**Table 4 Comparisons of measurements of the lower canines between the *E. jubatus* and GKZ-N 00001.**

|  | N | Min–Max | Mn |
|---|---|---|---|
| *E. jubatus* : Male (*Kohno & Tomida, 1993*) |  |  |  |
| MR (9+) (mm) | 4 | 38.6-42.4 | 39.1 |
| BR (9+) (mm) | 4 | 21.4-22.6 | 21.8 |
| BR/MR × 100 (%) | 4 | 51.6-61.8 | 56.6 |
| GKZ-N 00001 (this study) |  |  |  |
| MR (mm) | 1 | 50.3 |  |
| BR (mm) | 1 | 27.6 |  |
| BR/MR × 100 (%) | 1 | 54.9 |  |

**Note:**
MR: greatest mesiodistal diameter of the root BR: greatest buccolingual diameter of the root (9+) = over 9 years old (age determination is based on *Kubota et al., 1961*).

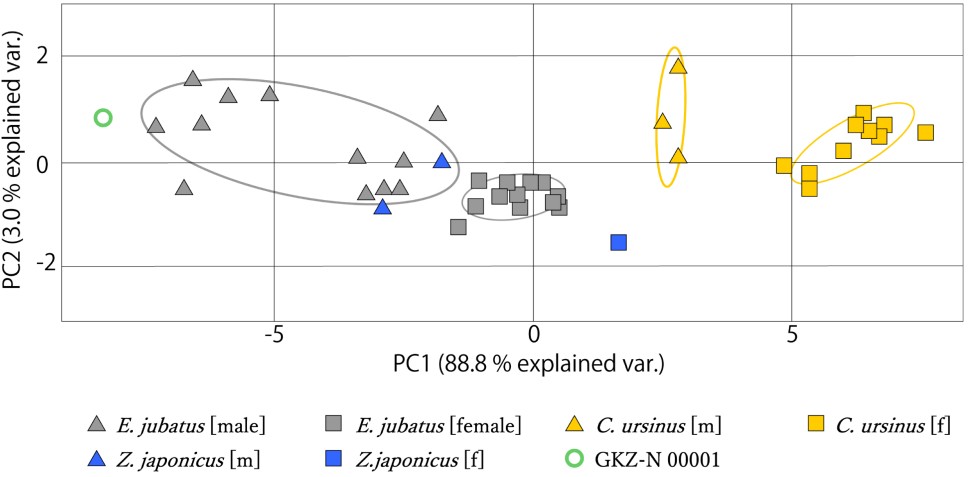

**Figure 6 Mandibular PCA results comparing PC1 and PC2.** Ovals represent 95% confidence intervals for each group (species and sex).

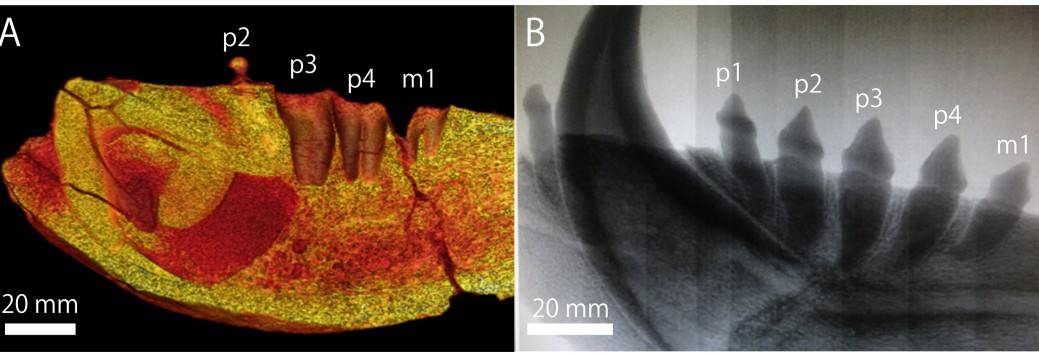

**Figure 7 Postcanine roots by CT image.** (A) GKZ-N 00001 (B) *E. jubatus* (NMNS-KK 14). Photo credits: (A) Naoki Kohno (B) Chisako Sakata.

the PC2 value, the longer the mandible anteroposteriorly. It also means that the higher the PC2 value, the deeper the mandible dorsoventrally.

## DISCUSSION

GKZ-N 00001 has lanceolate, simplified cheek teeth, and lacks m2. This condition is recognized to be a synapomorphy of the crown Otariidae (*King, 1983*; *Boessenecker & Churchill, 2015*; *Velez-Juarbe, 2017*). In addition, the overall size of this mandible is included within the range of those in the largest sea lion; that is, the Steller sea lion *E. jubatus*. Also a result of morphometric analyses for the size and proportion of the mandible for GKZ-N 00001 with extant species of a fur seal and sea lions such as the Northern fur seal, the Japanese sea lion and the Steller sea lion distributed in the western North Pacific suggest that GKZ-N 00001 is included in a range of *E. jubatus* and that it is inferred to be a male individual. Therefore, GKZ-N 00001 is distinctively identifiable to the extant species of the Steller sea lion *E. jubatus*. Although GKZ-N 00001 is slightly larger than male individuals sampled from the Recent population, it is at the moment uncertain whether this is simply the size variation among individuals of the same species or a potential implication of larger average size in the early Pleistocene population of the same species. At present, we consider that GKZ-N 00001 was just a large individual of *E. jubatus* until the multiple specimens of mandibles are obtained from the same and/or younger geochronologic formations.

Apart from the size difference, all the cheek teeth of the extant specimens of *E. jubatus* are single or bilobate single rooted. However, the detached isolated p4 of GKZ-N 00001 is bilobate single rooted in condition, and the alveolus for m1 is strongly bilobate and apically double rooted (Fig. 7; Fig. S1). Although this condition is not seen in the variation of extant specimens of *E. jubatus*, m1 root is always bilobate and its degree is strongly variable and successive among them (Fig. S1). Because the otariids (and stem taxa of odobenids in the Otarioidea) have a process that the cheek tooth condition is gradually changed from double rooted to single rooted to be homodont dentition during their evolution (*Berta & Deméré, 1986*; *Boessenecker, 2011*; *Velez-Juarbe, 2018*), the bilobate condition of p4 and the apically double rooted condition of m1 on GKZ-N 00001 are considered to be plesiomorphic conditions in a series of transition to the homodonty within

a same species as a parallel phenomenon during their evolution. Accordingly, the condition of the cheek tooth roots on GKZ-N 00001 would also simply be a primitive feature for the lineage of the Steller sea lion.

Finally, the recognition of GKZ-N 00001 as the oldest record of *Eumetopias jubatus* could also be important also for the molecular phylogenetics (see also *Parham et al., 2012*). Because only the subfamilial divergence estimates among the pinnipeds had calibrated based on fossil records of crown pinnipeds (*Yonezawa, Kohno & Hasegawa, 2009*), and because the oldest fossil record of *Eumetopias* had not been traced back only until the late Pleistocene, it could be considered the geologic age of GKZ-N 00001 as a good calibration point to improve divergence estimates between *Eumetopias* and its closest genus *Zalophus* and also of the crown Otariidae.

## CONCLUSIONS

The mandibular fossil (GKZ-N 00001) from the lower Pleistocene Omma Formation (0.8 Ma) is specifically identified as *E. jubatus* based on the morphometric analyses. Previously, the oldest record of *E. jubatus* was from the upper Pleistocene of North America (*Kohl, 1974*; *Whitmore & Gard, 1977*; *Barnes, Ray & Koretsky, 2006*), so GKZ-N 00001 from the upper lower Pleistocene (ca. 0.8 Ma) is now recognized as the oldest record of *E. jubatus*. This implies that the distribution of the Steller sea lion was already established at about by 0.8 Ma in the western North Pacific and at least southerly in the Sea of Japan.

## INSTITUTIONAL ABBREVIATIONS

**DCIFC**   Daté City Institute of Funkawan Culture, Daté City, Hokkaido, Japan
**GKZ-N**   Geological Collection, Kanazawa University, Kanazawa, New series, Japan
**HM**   Hokkaido Museum, Sapporo, Hokkaido, Japan
**NMNS-KK**   The Late Kinjiro Kubota Collection, Department of Geology and Paleontology, National Museum of Nature and Science, Tsukuba, Japan
**NSMT-M**   Mammalogical Collections, Department of Zoology, National Museum of Nature and Science, Tsukuba, Japan

## ACKNOWLEDGEMENTS

We thank Robert G. Jenkins (Kanazawa University), Yuko Tajima (National Museum of Nature and Science), Hiroshi Usiro and Yuji Soeda (Hokkaido Museum), Tomoya Aono (then Daté City Institute of Funkawan Culture, now Tohoku University of Art and Design), Yukihito Nagaya (Daté City Institute of Funkawan Culture), and Gen Nakamura (Tokyo University of Marine Science and Technology, Tokyo, Japan) for permitting us to use their collections and providing us with working space at respective institutes. We also thank Chisako Sakata (NMNS) for CT scanning specimens for this study. We are grateful to Katsuo Sashida (then University of Tsukuba, now Mahidol University), Sachiko Agematsu and Kohei Tanaka (University of Tsukuba), and Yasunari Shigeta (National Museum of Nature and Science, and University of Tsukuba) for their useful advice, discussion and generous encouragement during the course of this study.

### Funding

The authors received no funding for this work.

### Competing Interests

The authors declare that they have no competing interests.

### Author Contributions

- Nahoko Tsuzuku performed the experiments, analyzed the data, prepared figures and/or tables, and approved the final draft.
- Naoki Kohno conceived and designed the experiments, authored or reviewed drafts of the paper, and approved the final draft.

### Data Availability

The raw measurements are available in Table S1. The accession numbers of male *Eumetopias jubatus* specimens measured in this study are available in Data S1.

### Supplemental Information

Supplemental information for this article can be found online at http://dx.doi.org/10.7717/peerj.9709#supplemental-information.

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
