# Peer review of "The oldest record of the Steller sea lion Eumetopias jubatus (Schreber, 1776) from the early Pleistocene of the North Pacific"

_PeerJ, doi:10.7717/peerj.9709_

## Round 0.1 · original submission · Minor Revisions

The omission of Proterozetes needs to be rectified, as indicated by both reviewers, especially because this might impact the identification of your specimen.

·

Basic reporting

Main comments:

This new paper by Tsuzuku and Kohno redescribes a fossil mandible originally reported by Kaseno in the 1950s, and discussed a number of times since. Originally misidentified as the extinct pinniped Allodesmus, subsequent authors reassigned the specimen to the extant genus Eumetopias. This study redescribes and statistically evaluates this specimen, and the authors conclude that it is referable to Eumetopias jubatus. The paper is well-written and concise, and the statistical analyses are competently executed. I have a few points that need to be addressed before I can recommend acceptance.
1) This manuscript avoids mentioning the extinct sea lion Proterozetes. Proterozetes is coeval with this specimen, and recently reported from the Pleistocene of Japan in a conference presentation. Could this specimen belong to Proterozetes? Comparisons are needed to A) demonstrate that this specimen is not Proterozetes rather than Eumetopias and B) why Proterozetes was left out of the statistical analyses. On that note, what about Eumetopias ojiyaensis?
2) This specimen has double rooted teeth. Are single rooted teeth present in any extant specimens of E. jubatus? If not, this feature would actually fall outside the range of variation for the extant species and therefore would not be assignable to it. Pending the outcome of comparisons with Proterozetes, I think a strong case can be made for identification as Eumetopias. However, I have reservations over a species-level identification as E. jubatus. At best, I think such a dental difference could mean this may correspond to a chronospecies of Eumetopias ancestral to E. jubatus.
3) A more minor point, but some reference to other specimens of Eumetopias (e.g. Eumetopias ojiyaensis) would be useful.
Owing to these issues, I suggest major revision – and have every reason to believe these issues can be satisfactorily dealt with. Aside from these, I have several minor comments below:
Title: jubatus is misspelled.
Abstract: “Among then” should be “them”
Abstract: the age cited for the Omma Formation is incomplete, and should include the maximum age as well. Why not just state the precise age rather than rounding – e.g. 1.36-0.83 Ma?
39: “One such important fossil specimen has been recorded…”
53: recent should not be capitalized
60: “One such important…”
66: Awkward phrasing; when using the word yields, something yields something else, rather than something being yielded by another. In other words, it is “yielded by” rather than “yielded from”, if that makes sense. I’m no grammarian, so I don’t really know how to explain this very well! Regardless, I suggest changing this (here, and elsewhere) to “It was derived” or “It originated from” or “It was collected from”.
76: “turned out to be early Pleistocene”
140: Be specific – cold temperate? Is there a paleotemperature estimate?
174: “In form” can be deleted.
196: “The crown is tall and conical…”
197: delete ‘also’
244: I think the term ‘deeper’ is more appropriate here than ‘higher’
248: use the word ‘lacks’ instead of ‘doesn’t have’

Kind regards, Robert W. Boessenecker

Experimental design

I think that the statistical analyses are executed well and the results are informative, however the study does suffer form the exclusion of the similarly sized (and closely related) sea lion Proterozetes.

Validity of the findings

See above.

·

Basic reporting

See general comments.

Experimental design

See general comments.

Validity of the findings

See general comments.

Additional comments

This is an interesting manuscript and will make quite an important addition to our current knowledge of crown Otariidae. I commend the authors on the informative figures and descriptive text. I have carefully read the manuscript and have made some minor comments and stylistic suggestions as specified below, and there are two main issues which I would like to bring attention to. I hope the authors consider these recommendations and suggestions.

Major comments
Given the age and morphology, it would be advisable to do a thorough comparison with the mandibles of Proterozetes ulysses described by Poust and Boessenecker (2017) as this is relevant to the identification of GKZ-N 00001.
Being the oldest record of Eumetopias jubatus it could be considered as a calibration point that could help improve divergence estimates in crown Otariidae. This should be added to the discussion part of the manuscript. See for example, Parham et al. (2012), Syst. Biol. 61(2):346–359, and other related papers on this matter.

Minor comments.
Title has a typo: Eumetopias jubtus should be changed to Eumetopias jubatus
Line 36: “then,” should be changed to “them”
Line 58: I suggest changing “... that is now only around Hokkaido…” to “... which is now only around Hokkaido…”
Lines 59-60: I suggest changing “... relatively wide, because their mandibular and tooth fossils have…” to “... relatively broad, as dental and mandibular remains have…”
Lines 66-67: I suggest changing “It was yielded from…” to “The specimen came from…”
Line 69: I suggest changing “... that was an extinct…” to “... which is an extinct…”
Lines 82-83: I suggest changing “Therefore, exact identification and classification of GKZ-N 00001 is meaningful.” to “Therefore, highlighting the meaningfulness of proper identification of GKZ-N 00001.”
Line 108: Typo, reference “Veletz-Juarbe, 2017” should be “Velez-Juarbe, 2017”
Lines 119-123; Emended Diagnosis: I suggest the emended diagnosis should be more descriptive. As in, is there a proportion that can help illustrate the large size of the canines? For example, what is the diameter of the canines relative to the width of the rostrum or relative to the postcanine toothrow length? Is there a numerical range that can illustrate the large angle between the horizontal and ascending rami?
Line 137: I suggest changing “... Diplodonta are collected…” to “... Diplodonta have been collected…”
Line 140: “cold water” is there a known range of degrees that can help illustrate this better?
Line 163: I suggest changing “... and is slot shape.” to “... and is slit-like.”
Line 167: I suggest changing “... and rounded in form. It is 11 mm in diameter.” to “... and rounded in outline, with a diameter of 11 mm.”
Line 170: I suggest changing “... and is rounded in form. Its diameter is 6 mm.” to “... with a rounded outline and 6 mm in diameter.”
Line 171: I suggest combining the two sentences as in line 170.
Line 172: Same as above.
Line 174: Same as above.
Line 179: I suggest changing “... foramen is large. Its diameter is 7 mm.” to “ … foramen is large, with a diameter of 7 mm.”
Line 190: is the accessory cusp of p2 located mesially, distally or more central?
Line 191: I suggest changing “... bilobed in form.” to “... bilobed in outline.”
Line 248: Typo, change “laceorate” to “lanceolate”
Line 265: The first part of this sentence needs to be more clear, as it is, it can be misinterpreted as if otariid dentition would change from double to single rooted condition during ontogeny. It should be clarified that it has been shown to occur within a genus (i.e. Callorhinus) and more broadly, across other pinniped groups, although not necessarily in the same order as in otariids (see Velez-Juarbe, 2018, J. Vertebr. Paleontol. 38(4):e1481080).
Line 268: I suggest changing “... to be a primitive conditions in…” to “... to be a plesiomorphic condition in…”
Line 270: I suggest changing “... primitive…” to “... plesiomorphic…”
Line 276: I suggest changing “... jubatus has been known from…” to “... jubatus was from the upper…”
Figure 1: In figure 1B the digastric prominence is labeled as positioned on the posterior surface of the symphysis. This needs to be changed. In pinnipeds, the digastric muscle is “single-bellied” with its origin at the paroccipital processes, while its insertion is along the posteroventral edge of the mandibular ramus (see Howell, 1929). Also, see for example, the label of the digastric insertion in fig. 3 of Boessenecker (2011).

---

## Round 0.2 · accepted · Accept

I think this study will be important as a comparative study across the sealions of the northern hemisphere, especially with your quantification of many characters.